# A Novel Lightweight Model for Underwater Image Enhancement

**DOI:** 10.3390/s24103070

**Published:** 2024-05-11

**Authors:** Botao Liu, Yimin Yang, Ming Zhao, Min Hu

**Affiliations:** 1School of Computer Science, Yangtze University, Jingzhou 434025, China; liubotao920@yangtzeu.edu.cn (B.L.); mingzhao@yangtzeu.cn (M.Z.); 2Western Research Institute, Yangtze University, Karamay 834000, China

**Keywords:** structural reparameterization, channel attention mechanism, underwater image enhancement, feature extraction, feature fusion

## Abstract

Underwater images suffer from low contrast and color distortion. In order to improve the quality of underwater images and reduce storage and computational resources, this paper proposes a lightweight model Rep-UWnet to enhance underwater images. The model consists of a fully connected convolutional network and three densely connected RepConv blocks in series, with the input images connected to the output of each block with a Skip connection. First, the original underwater image is subjected to feature extraction by the SimSPPF module and is processed through feature summation with the original one to be produced as the input image. Then, the first convolutional layer with a kernel size of 3 × 3, generates 64 feature maps, and the multi-scale hybrid convolutional attention module enhances the useful features by reweighting the features of different channels. Second, three RepConv blocks are connected to reduce the number of parameters in extracting features and increase the test speed. Finally, a convolutional layer with 3 kernels generates enhanced underwater images. Our method reduces the number of parameters from 2.7 M to 0.45 M (around 83% reduction) but outperforms state-of-the-art algorithms by extensive experiments. Furthermore, we demonstrate our Rep-UWnet effectively improves high-level vision tasks like edge detection and single image depth estimation. This method not only surpasses the contrast method in objective quality, but also significantly improves the contrast, colorimetry, and clarity of underwater images in subjective quality.

## 1. Introduction

Underwater image enhancement, a technique used to restore underwater images to clarity, is a challenging task for the serious quality problems of underwater images in the water medium due to light absorption and scattering. Underwater images differ from normal image imaging in that different wavelengths of light have different energy attenuation rates during transmission. The longer the wavelength, the faster the attenuation rate. The red light with the longest wavelength decays faster, while the blue and green light decays relatively slowly, so the underwater images are mostly blue-green biased. In addition, this enhanced technology is heavily relied on by vision-guided robots and autonomous underwater vehicles to effectively observe regions of interest for a great deal of advanced computer vision tasks such as underwater docking [1], submarine cable and debris inspection [2], salient target detection [3], and other operational decisions. Therefore, how to solve the problems of color distortion, low contrast, and blurred details in underwater images is the main challenge for researchers today.

The solutions to these underwater image problems can be divided into two categories, one based on traditional enhancement methods and the other on deep learning methods [4].

The traditional methods can be divided into two categories, i.e., non-physical model-based enhancement methods [5] and physical model-based enhancement methods [6]. Firstly, non-physical model methods do not need to consider the imaging process, and such methods mainly include histogram equalization, grayscale world algorithm, Retinex algorithm, etc. The histogram equalization method [7] can evenly distribute the image pixels, which improves the image quality and sharpness to some extent. Grayscale world algorithm [8] removes the effect of ambient light from the image and enhances the underwater image. Fu et al. [9] proposed a Retinex-based underwater image enhancement method, applying the Retinex method to obtain the reflection and irradiation components based on the correction of the underwater image color, resulting in an enhanced underwater image. Ghani et al. [10] proposed Rayleigh stretched finite contrast adaptive histograms to normalize global and local contrast enhancement maps to enhance underwater image quality. Zhang et al. [11] used Retinex, bilateral filtering, and trilateral filtering in CIELAB color space for underwater image enhancement. Li et al. [12] used information loss minimization and histogram distribution to eliminate water fog and enhance the contrast and brightness of underwater images. In short, the above methods based on non-physical models have simple algorithms for fast implementation, but suffer from problems such as over-enhancement and artificial noise. Traditional image enhancement methods can, to a certain extent, eliminate image blur, enhance edges, etc. However, such methods improve the quality of underwater images only using a single image processing, by adjusting the image pixel values to improve the visual quality. Since the physical process of underwater image degradation is not taken into account, the achieved effect is limited and there are still problems such as high noise, low definition, and color distortion, so further enhancement and improvement are needed.

Considering these shortcomings, scholars have further proposed a physical model-based approach. The core idea is to construct a mathematical imaging model for the degradation process of underwater images. The parameters of the imaging model are estimated based on the observations and various a priori assumptions to derive an undegraded image in the ideal state. In the most classical dark channel prior (DCP) algorithm [13], researchers obtain light transmittance and atmospheric light estimates based on the relationship between the fog image and the imaging model, so as to achieve enhancement of the fog image. Since underwater images are somewhat similar to fogged images, the DCP algorithm is also used for underwater image enhancement, but its application scenario is very limited. Therefore, researchers proposed the Underwater Dark Channel (UDCP) algorithm [14] specifically for the underwater environment, which takes into account the attenuation characteristics of light underwater and estimates the transmittance of light waves in water more accurately to achieve underwater image enhancement. Peng et al. [15] proposed an underwater image restoration method to deal with underwater image blurring and light absorption, which introduces depth of field in the atmospheric scattering model and applies a dark channel a priori to solve for more accurate transmittance, thus achieving underwater image enhancement. To sum up, the methods based on physical models rely on imaging models and a priori knowledge of the dark channel [16], but the specificity of the underwater environment leads to the limitations of the methods. The physical model-based approach takes into account the optical properties of underwater images, but usually relies on environmental assumptions and specialized a priori knowledge of physics, and therefore has significant limitations. The estimation methods of model parameters are difficult to generalize to different underwater conditions and lack strong generalization and applicability.

In recent years, deep learning has attracted widespread attention with its remedy for the shortcomings of traditional methods [17]. The deep learning approach can reduce the impact of the complex underwater environment on the image to achieve better enhancement results. Both convolutional neural network (CNN)-based [18] models and generative adversarial network (GAN)-based [19] models require a large number of paired or unpaired datasets. Chen et al. [20] proposed an underwater image enhancement method that fuses deep learning with an imaging model, which obtains an enhanced underwater image by estimating the background scattered light and combining it with an imaging model for convolution operations. Islam et al. [21] proposed a fast underwater image enhancement model (FUnIE-GAN), which establishes a new loss function to evaluate the perceptual quality of images. Fabbri et al. [22] proposed a generative adversarial network (UGAN)-based method that enhances the details of underwater images, but has ambiguous experimental results occur since Euclidean distance loss is applied. Wang et al. [23] (2021) proposed an unsupervised underwater generative adversarial network (underwater GAN, UWGAN), which synthesize realistic underwater images (with color distortion and haze effects) from aerial images and color depth map pairs based on an improved underwater imaging model, and directly reconstructs clear images underwater based on the synthesized underwater dataset using an end-to-end network. In summary, the above underwater image enhancement algorithm based on deep learning improves the overall performance of the algorithm. The techniques in the existing literature are mainly based on very deep convolutional neural networks (CNNs) and generative adversarial network (GAN) models, focusing on noise removal for image defogging, contrast extension, combination with multi-information improvement and deep learning, etc. However, these large models require a high amount of computation and memory, which makes it difficult to perform real-time underwater image enhancement tasks.

In this paper, a lightweight model Rep-UWnet based on structural reparameterization (RepVGG) [24] is designed to recover underwater images by addressing the problems of color distortion, detailed features loss, large memory consumption, and high computation in the underwater images enhanced by existing algorithms. In addition, some ideas from Shallow-UWnet [25] are adopted in the model. Although Shallow-UWnet has a smaller number of parameters, its model accuracy and inference speed need to be further improved. In this paper, RepVGG Block is used instead of a normal convolution, which leads to an average inference speedup of 0.11 s. Secondly, a multi-scale hybrid convolutional attention module is designed in this paper, which leads to an improvement of model accuracy by about 11.1% in PSNR, 9.8% in SSIM, and 7.9% in UIQM. We also decrease the channel of convolutions to design the lightweight model, so the overall model has approximately 0.45 M parameters, which is less and faster than other state-of-the-art models. According to the experimental conclusion, the innovation points of the model in this paper are as follows.

(1)A multi-scale hybrid convolutional attention module is designed. Considering the complex and diverse local features of underwater scenes, this paper uses a spatial attention mechanism and channel attention mechanism. The former is to improve the network’s ability to pay attention to complex regions such as light field distribution and color depth information in underwater images, while the latter focuses on the network’s representation of important channels in features, thus improving the overall representation performance of the model.(2)RepVGG Block is used instead of ordinary convolution, and different network architectures are used in the network training and network inference phases. With more attention to accuracy in the training phase and more attention to speed in the inference phase, an average speedup of 0.11 s for a single image test is achieved.(3)A joint loss function combining content perception, mean square error, and structural similarity is designed, and weight coefficients are applied to reasonably assign each loss size. For the perceptual loss, layers 1, 3, 5, 9, and 13 of the VGG19 model are selected in this paper to extract hidden features and generate clearer underwater images while maintaining the original image texture and structure.

## 2. Related Work

### 2.1. Neural Net Feature Fusion

Neural network feature fusion is a key technique in the field of deep learning, aiming to integrate feature information from different layers or sources to improve the performance and generalization of models. Feature fusion becomes particularly important when dealing with challenging tasks such as underwater images, which often suffer from low contrast and color distortion. This research is dedicated to solving this problem by proposing a lightweight model, Rep-UWnet, which enhances the quality of underwater images through feature fusion techniques and improves the performance of the model while reducing storage and computational resources.

Classical deep learning models, such as Inception [26], ResNet [27], and DenseNet [28], provide many effective approaches in feature fusion. The Inception model summarizes feature mappings through different scales of convolution, ResNet introduces Skip connections to facilitate the propagation of gradients, and DenseNet enhances feature-to-feature connectivity through dense connections for the flow of information between features. However, these approaches tend to neglect the importance of each feature mapping, so attention mechanisms are introduced to address this problem.

SENet [29] and CBAM [30] are two typical attention mechanisms, both of which play an important role in feature fusion. SENet introduces a channel attention mechanism, which improves the model’s attention to important feature channels by recalibrating the channel feature responses. CBAM divides the attention mechanism into spatial attention and channel attention, which further improves the model’s attention to features. Additionally, the attention mechanism is not limited to feature fusion, but has also been widely applied in the field of image processing, such as tasks like image denoising [31], de-raining [32], and de-fogging [33].

In this study, we will combine the existing classical models and attention mechanisms, and propose a multi-scale hybrid convolutional attention module to adaptively control the weights of feature mappings at different scales. This novel approach will further improve the effectiveness of underwater image enhancement and achieve better performance in terms of model performance and robustness.

### 2.2. Underwater Datasets

In this paper, three underwater image datasets are collected to be trained and tested on EUVP [21] and UFO 120 [34] datasets, and compared with different models to demonstrate the performances and generalization capabilities of the proposed model. Detailed data are shown in Table 1 below.

(1)The EUVP (Enhancing Underwater Visual Perception) dataset is a large dataset designed to facilitate supervised training of underwater image enhancement models. The dataset contains both paired and unpaired image samples, covering images with poor and good perceptual quality for model training. The EUVP dataset consists of three sub-datasets, namely, Underwater Dark, Underwater ImageNet, and Underwater Scene, which cover different types of waters and underwater landscapes, such as oceans, lakes, rivers, coral reefs, rocky terrains, and seagrass beds. These images exhibit rich diversity and representativeness, reflecting the typical characteristics of real-world underwater environments and covering waters in different geographic regions. In the process of acquiring the EUVP dataset, the researchers fully considered the effects of factors such as water quality, lighting conditions, and shooting equipment on image quality and model performance. The dataset utilized several different types of cameras, including GoPro cameras, Aqua AUV’s uEye cameras, low-light USB cameras, and Trident ROV’s HD cameras, to capture image samples under different conditions. These data were collected during ocean exploration and human–computer collaboration experiments in various locations and under different visibility conditions, including images extracted from a number of publicly available YouTube videos. The images in the EUVP dataset have therefore been carefully selected to accommodate the wide range of natural variability in the data, such as scene, water body type, lighting conditions, and so on. By controlling these factors, the quality and reliability of the data are ensured, providing an important database for model training.(2)The UFO 120 dataset, comprising 1500 pairs of underwater images, serves as a pivotal resource aimed at bolstering the training of algorithms and models for underwater image processing. Despite its relatively modest size, this dataset encapsulates a diverse array of scenes, water body types, and lighting conditions, showcasing a representative cross-section of underwater environments. Sourced from the Flickr platform, these images authentically capture the myriad complexities present in real-world underwater settings. Throughout the data collection process, meticulous consideration was given to factors such as water quality, lighting variations, and equipment configurations, ensuring the portrayal of a wide spectrum of visual characteristics and challenges. Serving as a real-world benchmark, the UFO 120 dataset imposes rigorous demands on underwater image processing models, necessitating their adeptness in handling diverse lighting scenarios and water conditions, while also addressing potential issues like noise and blurriness. Consequently, conducting evaluations and tests using the UFO 120 dataset facilitates a comprehensive appraisal of a model’s real-world performance, thereby fostering avenues for refinement and optimization. Despite its relatively modest size, the UFO 120 dataset holds significant research value, furnishing invaluable insights and benchmarks for advancing research in the realm of underwater image processing.

**Table 1 sensors-24-03070-t001:** Details of the number and size of datasets.

Dataset Name	Training Pairs	Validation	Total Images	Size
Underwater Dark	5550	570	11,670	256 × 256
Underwater ImageNet	3700	1270	8670	256 × 256
Underwater Scenes	2185	130	4500	256 × 256
UFO 120	1500	120	3120	320 × 240

## 3. Proposed Method

### 3.1. Rep-UWnet Model

Figure 1 shows the schematic architecture of the proposed Rep-UWnet model. The original image for the input of the model in this paper is a 256 × 256 RGB underwater low-quality image. First, after being extracted by the SimSPPF module and added with the original image features to obtain the image multi-scale information and enhance the perceptual field, the features are then used as the input image. Second, the input image is connected to the output of each RepConv block through a Skip connection. The input image goes through the first convolutional layer with a kernel size of 3 × 3 to generate 64 feature maps, and then the multi-scale hybrid convolutional attention module is used to enhance the useful features by reweighting the features of different channels. Third, three RepConv blocks are linked to reduce the number of parameters while extracting features, so as to increase the test speed. Finally, a convolutional layer with 3 kernels generates the enhanced underwater image.

RepConv: This part consists of two sets of RepVGG blocks and a set of convolution blocks. Each RepVGG block is followed by a Dropout [35] and SiLU activation function, which help to prevent overfitting. Then, the output is passed through another set of Conv-SiLU pairs, which helps to stitch the input images from the Skip connections. RepVGG makes it possible to use different network architectures for the network training and network inference phases, with the training phase being more concerned with accuracy and the inference phase being more concerned with speed.

Skip connections: The original input image is connected to the output of each remaining block by Skip connections. Traditional convolutional neural network models increase the color depth of the network by stacking convolutional layers, thus increasing the recognition accuracy of the model. However, when the network level increases to a certain number, the accuracy of the model decreases because the neural network is propagating backwards. During the process, the propagation gradient is required to be continuous, but it will fade out as the number of network layers deepens, making it impossible to adjust the weights of previous network layers. In this paper, Skip connections ensure feature learning from each block, as well as the basic features from the underlying original image, which prevents the network from overfitting to the training data, thus investing the model with a strong generalization capability.

SimSPPF module [36]: SimSPPF is a spatial pyramidal pooling structure throughout, mainly to solve two problems. The first is the problem of image distortion caused by cropping and scaling operations on image regions; the second is the problem of repeated feature extraction of graph correlation by convolutional neural networks, which greatly influences the speed of producing clear images and the computational costs.

### 3.2. Multi-Scale Hybrid Convolutional Attention Module

To further enhance the performance of Rep-UWnet, this paper introduces a critical multi-scale hybrid convolutional attention module, as illustrated in Figure 2a. This module plays a pivotal role in underwater image enhancement. Considering the complexity and diversity inherent in underwater scenes, we initially employ 1 × 1 and 3 × 3 convolutional operations to obtain varying receptive fields for multi-feature fusion. Additionally, we utilize residual concatenation to address the issue of gradient vanishing, ensuring that the spatial structure and color integrity of specific regions within the underwater image remain unaffected by scene quality. Subsequently, we augment the model’s capacity to capture and represent key details in underwater imagery by integrating spatial attention (Figure 2b) and channel attention modules (Figure 2c). Specifically, the multi-scale hybrid convolutional attention module aids the model in better perceiving and leveraging image features across different scales, thereby significantly enhancing the model’s performance and robustness.

### 3.3. RepVGG Block

RepVGG has influential work in the backbone, and its core principle is as follows: through the application of structural reparameterization, the training network of multi-way structure (the advantage of multi-branch model training—high performance) is transformed into the inference network of single-way structure (the benefit of model inference—fast, memory-saving) with the structure of 3 × 3 convolutional kernel; at the same time, the computational library (such as CuDNN, Intel MKL) and hardware for 3 × 3 convolution are deeply optimized to achieve the efficient inference rate of the network in the final. The structure of RepVGG’s block during training consists of three branches: 3 × 3 convolution, 1 × 1 convolution, and identity mapping. When downsampling, the above structure is adjusted as the convolution stride is changed into 2 and the identity mapping branch is removed. The specific structure is shown in Figure 3 below, where Figure 3a is the RepVGG Block structure during downsampling (stride = 2), and Figure 3b is the normal (stride = 1) RepVGG Block structure. Figure 3b shows that the RepVGG Block is trained with three branches in parallel: a main branch with the convolutional kernel size of 3 × 3, a shortcut branch with a convolutional kernel size of 1 × 1, and a shortcut branch with only BN attached. The trained RepVGG Block is converted into the model structure at the time of inference, i.e., the structural re-parameterization technique process. According to Figure 3c, it can be seen that the structural re-parameterization is mainly divided into two steps: the first step is mainly to fuse the Conv2d operator and BN operator as well as to convert the branch with only BN into one Conv2d operator, and the second step is to fuse the 3 × 3 convolutional layers on each branch into one convolutional layer.

### 3.4. Loss Function

To train the Rep-UWnet model in this paper, the following three loss functions are used in this paper: $L_{VGG}$, $L_{MSE}$, $L_{SSIM}$. First, $L_{VGG}$ is the perceptual loss, which utilizes the feature layer extracted from the pre-trained VGG19 model [37] as a loss network, with the aim of maintaining the consistency of the perceptual structure. Let $\phi_{j}$(x) be the jth post-activation convolutional layer of the pre-trained VGG19 network. The content loss is expressed as the difference between the feature representation of the enhanced image $k_{i}$ and the reference image $k_{i}^{*}$. The following Equation (1) is shown as:(1)$$L_{VGG}=\frac{1}{C_{j}H_{j}W_{j}}\overset{N}{\underset{i=1}{\sum }}\|\phi_{j}\left(k_{i}\right)-\phi_{j}\left(k_{i}^{*}\right)\|$$
where N is the number of each batch in the training process; $C_{j}H_{j}W_{j}$ denotes the dimension of the feature map of the jth convolutional layer within the VGG19 network. $C_{j}$,$H_{j}$ and $W_{j}$ are the number, height, and width of the feature maps, respectively.

$L_{MSE}$ is the mean squared error loss: MSE is a convenient way to measure the “mean error”, which evaluates the degree of variability of the data. The smaller the value of MSE, the better the accuracy of the prediction model in describing the experimental data. The sum of squares of the difference between the enhanced image k and the reference image $k^{*}$ is calculated from the mean squared error loss (MSE) of the pixels:(2)$$L_{MSE}=\frac{1}{N}\overset{N}{\underset{i=1}{\sum }}{\left(k_{i}-k_{i}^{*}\right)}^{2}$$

$L_{SSIM}$ is the structural similarity index (SSIM), representing the SSIM loss between the feature representation of the enhanced image and the reference image, calculated as:(3)$$SSIM\left(k_{i},k_{i}^{*}\right)=\frac{\left(2\mu_{x}\mu_{y}+C_{1}\right)\left(2\sigma_{xy}+C_{2}\right)}{\left(\mu_{x}^{2}+\mu_{y}^{2}+C_{1}\right)\left(\sigma_{x}^{2}+\sigma_{y}^{2}+C_{2}\right)}$$
where µ and σ denote the mean, standard deviation, and covariance of the image, and $C_{1}$ and $C_{2}$ are variables for stable division. The loss function of SSIM can be written as:(4)$$L_{SSIM}=1-SSIM\left(k_{i},k_{i}^{*}\right)$$

Finally, the content perception loss, mean square error (MSE) loss, and structural similarity (SSIM) loss are weighted together, and the loss function is defined as:(5)$$L_{Total}=\lambda_{1}L_{VGG}+\lambda_{2}L_{MSE}+\lambda_{3}L_{SSIM}$$
where $\lambda_{1}$, $\lambda_{2}$, and $\lambda_{3}$ are the weight indices to adjust the size of each loss. In the training, their values are used as hyperparameters for tuning.

## 4. Experiments

### 4.1. Dataset and Experimental Setup

Dataset. A total of 3000 paired images on EUVP underwater image are selected for training. Due to its diversity of capture locations and perceptual quality, the EUVP dataset is chosen as the training dataset, so that the model in this paper can be generalized to other underwater datasets. In addition, 515 paired test samples on EUVP and 120 pairs of test sets from the UFO 120 dataset are selected for testing.

Training setup. First, during the training period, the images are scaled to 256 × 256. Second, for perceptual loss, layers 1, 3, 5, 9, and 13 in the VGG19 model are chosen to extract hidden features. Third, λ_1_, λ_2_, and λ_3_ are set to 1, 0.6, and 1.1, respectively, in the experiments. Fourth, the Adam optimizer is applied to iterate through 200 rounds with a learning rate set to 0.0002 and batch-size of 4. Fifth, for the platform, the experiments are conducted on the Pytorch 2.3 with a CPU intel (R) Core(TM) i7-10870H CPU @ 2.20 GHz (Santa Clara, CA, USA), 16 GB of running memory, and NVIDIA GeForce RTX 2080Ti GPU 11 GB (Santa Clara, CA, USA) for training and testing. Sixth, the network training time is about 10 h.

Evaluation indicators. Three evaluation indicators are used to analyze the quality of the generated output images, including peak signal-to-noise ratio (PSNR), structural similarity index measure (SSIM), and reference-free underwater image quality measure (UIQM). Peak signal-to-noise ratio (PSNR) is used to express the ratio between the maximum possible power of a signal and the power of the corrupted noise that affects the fidelity of its representation. Since many signals have a very wide dynamic range, PSNR is often represented as a logarithmic quantity with a decibel scale. In image processing, it is primarily used to quantify the reconstruction quality of images and videos affected by lossy compression, and it is often defined simply by the mean squared error (MSE). The PSNR metric is given by the following equation:(6)$$PSNR\left(x,y\right)=\log_{10}\left(\frac{255^{2}}{MSE\left(x,y\right)}\right)$$

The structural similarity index (SSIM) is an index used to measure the similarity between two digital images. When two images are taken, one without distortion and the other after distortion, the structural similarity of the two images can be considered as a measure of the image quality of the distorted image. Compared to traditional image quality measures, such as peak signal-to-noise ratio (PSNR), the structural similarity is more consistent with the human eye’s judgment of image quality. SSIM is defined as Equation (3) above.

Unreferenced underwater image quality (UIQM) consists of three underwater image attribute indicators: image colorimetry (UICM), sharpness (UISM), and contrast (UIConM), where each attribute evaluates one aspect of underwater image degradation. The UIQM is given by the following equation:(7)$$UIQM=a_{1}\times UICM+a_{2}\times UISM+a_{3}\times UIConM$$
where the parameters $a_{1}=0.028$, $a_{2}=0.295,\;a_{3}=3.575$ are set according to the (Panetta, Gao, and Agaian [38]) paper. In addition, this paper measures the model compression and acceleration performance with compression rates and acceleration rates:(8)$$Compression\;rate\;\left(N,N^{*}\right)=\frac{\alpha \left(N\right)}{\alpha \left(N^{*}\right)}$$
(9)$$\mathrm{Speed}-\mathrm{up}\;\mathrm{rate}\;\left(N,N^{*}\right)=\frac{\beta \left(N\right)}{\beta \left(N^{*}\right)}$$
where $\alpha \left(N\right)$ is the number of parameters in model N, $\beta \left(N\right)$ is the test time per image in model N, N is the original model, and $N^{*}$ is the compressed model.

### 4.2. Experimental Results

The main reason for comparing our method with CLAHE, DCP, HE, ILBA, UDCP, Deep SESR, FUnIE-GAN, and U-GAN is that they represent a range of techniques in the field of underwater image enhancement. CLAHE is a classic technique for enhancing image contrast, but it may introduce artifacts and retain a haze-like effect. DCP, based on dark channel prior, performs well in dehazing and image enhancement but may generate artifacts in complex scenes. HE is a simple and intuitive method for image enhancement but has limited effectiveness in handling images with high noise or uneven contrast. ILBA, an improved method for uneven lighting conditions, can enhance image contrast and details but may fail in complex scenes. UDCP, a dark channel prior method specifically for underwater images, exhibits some robustness but may have issues with high-contrast and multimodal images. Deep SESR, a deep learning-based super-resolution method, has good performance and generalization, but requires a large amount of training data and computational resources. FUnIE-GAN and U-GAN, two generative adversarial network-based methods, enhance image clarity and contrast but require longer training times and significant computational resources. Through in-depth analysis of these methods, our study gains a better understanding of their characteristics, strengths, and limitations, providing targeted directions for improvement and optimization of our proposed underwater image enhancement algorithm.

In this section, our proposed method is compared subjectively and objectively with CLAHE [11], DCP [13], HE [6], UDCP [22], ILBA [15], U-GAN [14], FUnIE-GAN [21], and Deep SESR [39], representing various underwater image enhancement algorithms. CLAHE enhances image contrast but may introduce artifacts and retain a haze-like effect. DCP has limited effectiveness in improving the quality of underwater images. HE improves image quality, but there are some limitations. UDCP and ILBA have limited effectiveness in recovering images with specific color tones. U-GAN, Deep SESR, and FUnIE-GAN enhance image contrast but may have limitations in color recovery and artifact avoidance. In contrast, our proposed method not only effectively enhances the recovery of underwater images, but also improves features such as color bias and low contrast, resulting in more natural and clear images. Additionally, our method achieves better subjective quality, closer to the reference image (as shown in Figure 4j). Figure 4 displays some images enhanced by these methods, among which our method demonstrates results closer to the reference image in Figure 4j.

As described in 4.1, the peak signal-to-noise ratio (PSNR), structural similarity index measure (SSIM), and underwater image quality measure (UIQM) [38] are chosen as objective indicators for quantitative evaluation. The larger the indicator values are, the better the images generated. The results are shown in Table 2: the proposed method outperforms all the algorithms in terms of the indicators on the EUVP dataset, but it only achieves the second-best results on the UFO 120 dataset, which is probably because the best performer Deep SESR is trained on UFO 120.

The experiment employed a subjective evaluation grading standard to comprehensively assess the effectiveness of various underwater image enhancement algorithms. The grading standard consisted of five levels, ranging from “very poor” to “very good,” to describe the overall image quality and its impact on visual experience. By inviting five students to evaluate a range of representative algorithms, including CLAHE, DCP, HE, IBLA, UDCP, Deep SESR, FUnIE-GAN, U-GAN, and our proposed algorithm, we obtained comprehensive subjective evaluation data. Traditional underwater image processing algorithms (such as CLAHE, DCP, HE, IBLA, and UDCP) received relatively average scores in subjective evaluations, with average scores ranging from 2.8 to 3.2. In comparison, deep learning algorithms (such as Deep SESR, FUnIE-GAN, and U-GAN) achieved higher average scores, ranging from 3.8 to 4.0, demonstrating superior performance. Particularly, our algorithm obtained the highest average score of 4.0 in subjective evaluations, indicating significant advantages. Therefore, while traditional algorithms show some effectiveness in underwater image processing, deep learning algorithms perform better, with our algorithm exhibiting the best performance across all evaluations, providing superior visual effects for underwater image processing. The results are shown in Table 3.

In this paper, the RepVGG Block is replaced with a normal residual network, but this causes an increase in the number of parameters and a decrease in the testing speed. In addition, the proposed model has the lowest number of parameters and the shortest testing time for a single image compared to other deep learning-based models. This indicates that the structural reparameterization of RepVGG Block helps to speed up the network training and testing, resulting in an average speed increase of 0.11 s for single-image testing. The results are shown in Table 4.

### 4.3. Ablation Experiments

#### 4.3.1. Loss Function Ablation Experiment

In order to verify the effects of the mean square error loss term, structural similarity loss term, and content perception loss term in the loss function on the experimental results, ablation experiments are conducted on the above three loss terms on the EUVP dataset. In each ablation experiment, one of the loss terms is removed to conduct a comparative study.

From the subjective aspect, the image generated by the proposed method is closer to the reference image (Figure 5f), while the image generated with loss term removal suffers from obvious color bias, as shown in Figure 5. In the figure, w/o indicates the removal of a loss term in the loss function.

From the objective aspect, with the highest index of the complete method proposed in this paper, the loss terms are proven to be effective. The results of the objective quality comparison in ablation experiments are shown in Table 5.

#### 4.3.2. Attention Ablation Experiment

Another ablation experiment is also conducted to demonstrate the effectiveness of the multi-scale hybrid convolutional attention module. In this paper, two training methods are proposed as follows: (1) Rep-UWnet+multiscale hybrid convolutional block w/o spatial attention, and (2) Rep-UWnet+multiscale hybrid convolutional block w/o channel attention. In Table 6, it is demonstrated that multi-scale-based channel attention and spatial attention allow the proposed model to better learn the features of real underwater complex environments and obtain better indicators.

### 4.4. Application Testing Experiments

Rep-UWnet is a lightweight model suitable for various advanced visual tasks. This paper focuses on studying the problems of edge detection [40] and single-image color depth estimation [41] in the underwater environment. Underwater images often become blurred due to light attenuation and water quality effects, further reducing the accuracy of edge detection and single-image color depth estimation. To better investigate these issues, this paper selected the EUVP dark dataset, which is blurrier and allows for a more accurate evaluation of algorithm performance.

In this study, we observed that MiDaS [42] is affected by the green and blue tones in single-image color depth estimation, while HED [43] also faces similar issues in edge detection. As shown in Figure 6b, the color depth estimation contours are not clear in the original image, and the edge information in edge detection is insufficient. However, the performance of edge detection and color depth estimation is significantly improved in the enhanced images produced by our method, as illustrated in Figure 6d. This further demonstrates the effectiveness of our approach.

However, conducting edge detection and color depth estimation tasks in underwater environments also poses several challenges. Firstly, light attenuation and changes in water quality in underwater environments can lead to unstable image quality, which may affect the performance of models. Secondly, the deployment and optimization of underwater equipment are also challenging, as they may be affected by factors such as water flow, water pressure, and water temperature, which can affect image acquisition and sensor performance.

To overcome these challenges, this paper needs to design and optimize models tailored to the characteristics of underwater environments. For example, advanced image enhancement techniques can be employed to improve the quality of underwater images, thereby enhancing the accuracy of edge detection and color depth estimation. Additionally, the use of high-performance sensors and stable mechanical structures can improve the stability and reliability of underwater equipment, ensuring the effectiveness and reliability of the model in practical applications.

## 5. Conclusions

With this work, we successfully propose a lightweight underwater image enhancement method, Rep-UWnet, which aims to address the challenge of limited computational resources in underwater environments. Rep-UWnet employs a multi-scale hybrid convolutional attention module and combines spatial attention and channel attention mechanisms, which enables it to effectively enhance the network’s attention to complex regions of underwater images, such as light field distribution and color depth information. By using RepVGG Block instead of standard convolution, we successfully increase the average speed of single image tests and reduce the number of parameters of the overall model to about 0.45 M, thus outperforming other state-of-the-art models. The experimental results show that Rep-UWnet has superior performance compared to other models. In addition, we conducted ablation experiments to further demonstrate the effectiveness of each module. Due to its versatility and lightweight structure, Rep-UWnet can not only improve the performance of underwater image enhancement tasks, but also achieve significant results in advanced vision tasks such as edge detection and single image color depth estimation. In the future, we will continue to explore the potential of this method in other image enhancement tasks, such as image defogging and rain removal, to meet the needs of different robotic applications.

## Figures and Tables

**Figure 1 sensors-24-03070-f001:**
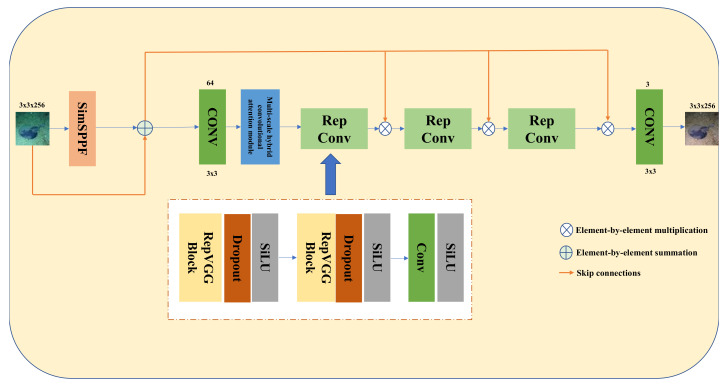
Structure of Rep-UWnet model.

**Figure 2 sensors-24-03070-f002:**
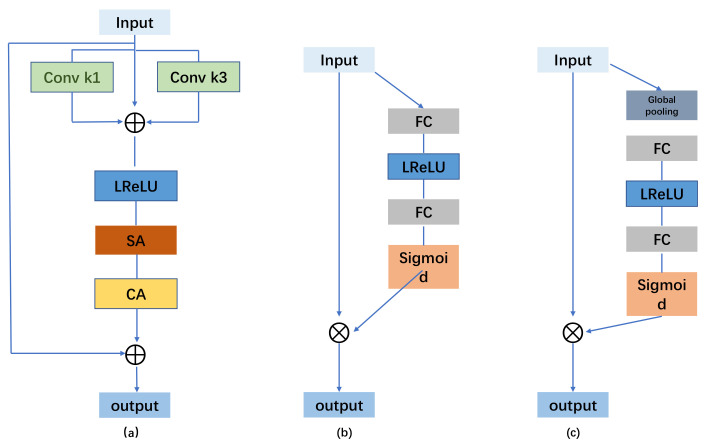
(**a**) The multiscale hybrid convolutional attention module; (**b**) spatial attention; and (**c**) channel attention are shown.

**Figure 3 sensors-24-03070-f003:**
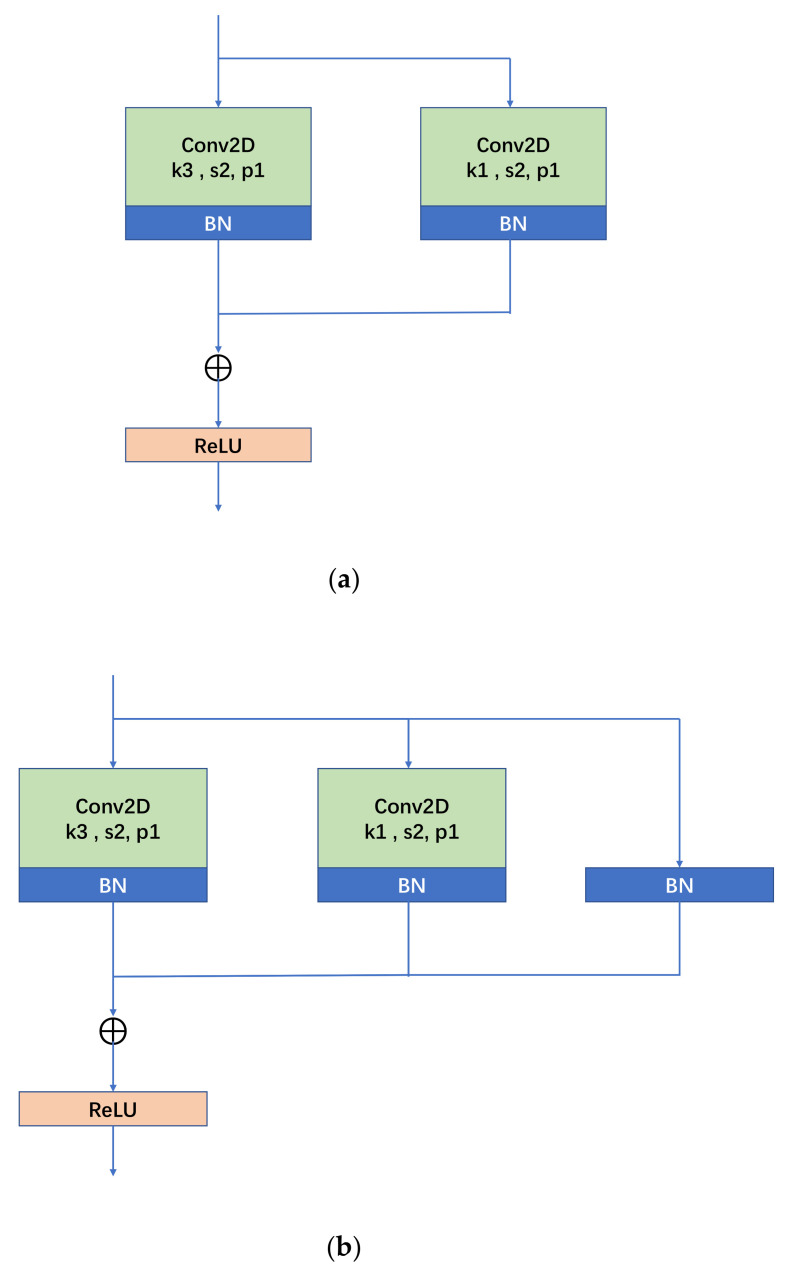
RepVGG Block structure, (**a**) down sampling; (**b**) normal; (**c**) structural re-parameterization process diagram.

**Figure 4 sensors-24-03070-f004:**
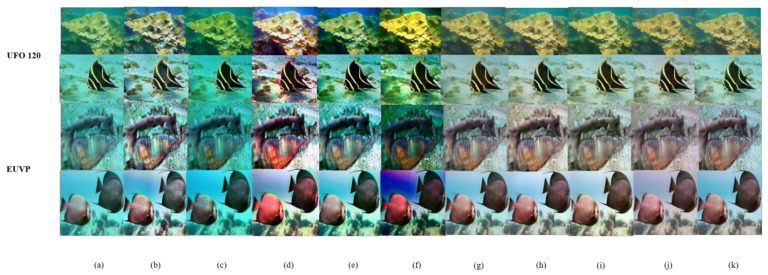
Subjective comparison of Rep-UWnet with existing methods and SOTA models for underwater image enhancement performance on EUVP and UFO 120 datasets. (**a**) input; (**b**) CLAHE [11]; (**c**) DCP [13]; (**d**) HE [6]; (**e**) ILBA [7]; (**f**) UDCP [6]; (**g**) Deep SESR [16]; (**h**) FUnIE-GAN [26]; (**i**) U-GAN [14]; (**j**) ours; (**k**) label.

**Figure 5 sensors-24-03070-f005:**
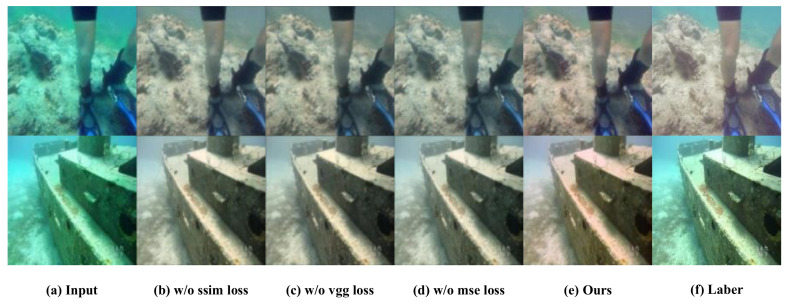
Ablation experiments on the EUVP dataset. “w/o” indicates that the removal of the corresponding loss term in the experiment. (**a**) Original underwater image (**b**) with SSIM loss removed, (**c**) with VGG loss removed, (**d**) with MSE loss removed, (**e**) image generated by the proposed methods, and (**f**) reference image.

**Figure 6 sensors-24-03070-f006:**
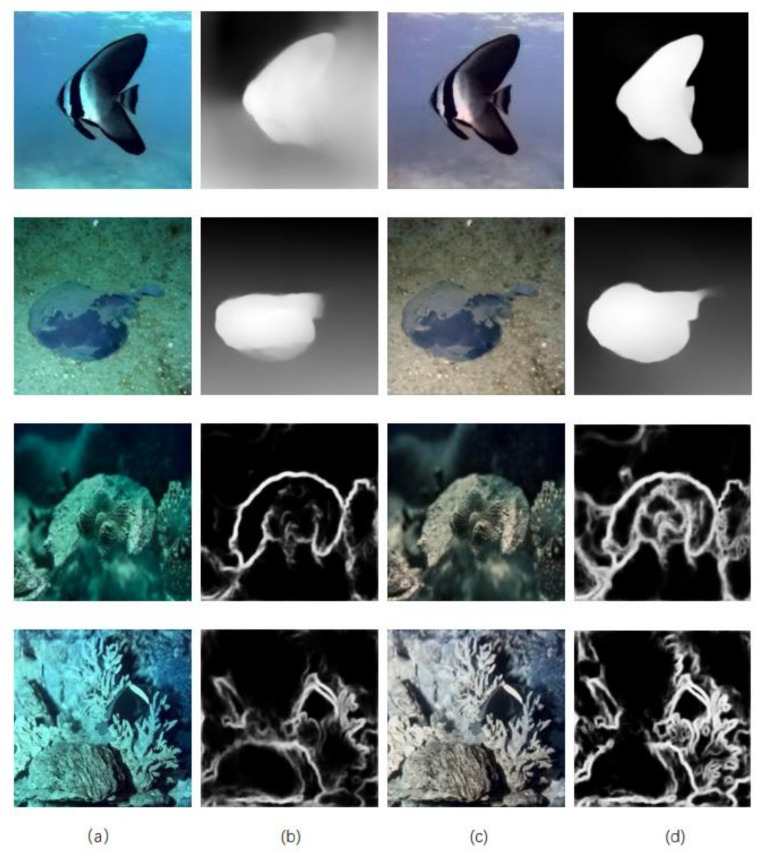
Example of EUVP dark dataset, single image depth estimation, and edge detection for real-world underwater images. (**a**) The original image, (**b**) is the result of single-image depth estimation and edge detection of (**a**), (**c**) is the image after enhancement by the method in this paper, and (**d**) is the result of single-image depth estimation and edge detection of (**c**).

**Table 2 sensors-24-03070-t002:** Mean PSNR, SSIM, and UIQM values of the enhanced results on the EUVP and UFO 120 datasets. The best two results are indicated in red and blue. PSNR, SSIM, and UIQM scores are expressed as mean $\pm \sqrt{variance}$.

Methods	UFO 120			EUVP		
	PSNR	SSIM	UIQM	PSNR	SSIM	UIQM
CLAHE	18.54 ± 2.08	0.69 ± 0.04	2.65 ± 0.38	19.31 ± 2.11	0.70 ± 0.11	2.55 ± 0.13
DCP	13.35 ± 1.77	0.67 ± 0.09	1.93 ± 0.27	13.66 ± 2.18	0.66 ± 0.03	1.88 ± 0.15
HE	16.08 ± 2.22	0.63 ± 0.07	1.88 ± 0.31	17.13 ± 1.66	0.64 ± 0.02	1.91 ± 0.16
IBLA	16.63 ± 2.13	0.64 ± 0.06	1.83 ± 0.24	17.22 ± 2.28	0.68 ± 0.05	1.88 ± 0.12
UDCP	16.25 ± 2.47	0.61 ± 0.03	1.66 ± 0.33	15.33 ± 2.64	0.65 ± 0.04	1.63 ± 0.18
Deep-SESR	26.46 ± 2.63	0.72 ± 0.05	2.93 ± 0.18	25.20 ± 2.26	0.78 ± 0.06	2.88 ± 0.23
FUnIE-GAN	24.72 ± 2.54	0.74 ± 0.06	2.79 ± 0.27	26.12 ± 2.83	0.81 ± 0.13	2.85 ± 0.27
UGAN	24.33 ± 1.37	0.70 ± 0.12	2.55 ± 0.24	23.65 ± 2.17	0.71 ± 0.02	2.83 ± 0.33
Ours	25.25 ± 2.44	0.73 ± 0.08	2.85 ± 0.19	27.50 ± 2.78	0.83 ± 0.14	2.93 ± 0.17

**Table 3 sensors-24-03070-t003:** Subjective evaluation scores of underwater image enhancement algorithms.

Algorithms	1	2	3	4	5	Average Score
CLAHE	3	3	3	4	3	3.2
DCP	2	3	3	2	3	2.6
HE	2	2	2	1	2	1.8
IBLA	3	3	3	3	3	3
UDCP	3	3	2	2	3	2.8
Deep-SESR	4	3	4	4	4	3.8
FUnIE-GAN	4	4	4	3	4	3.8
UGAN	4	3	4	3	4	3.6
Ours	4	4	5	3	4	4

**Table 4 sensors-24-03070-t004:** Model compression and acceleration performance indicators mentioned above. “w/o” indicates that the item is not added in the experiment.

Models	# Model Parameters	Compression Rate	Testing per Image (s)	Speed-Up
Deep-SESR	2.45 M	13.22	0.16	8
FUnIE-GAN	5.5 M	17.67	0.18	7
UGAN	38.7 M	55.34	1.13	24
Ours w/o RepVGG Block	1.23 M	1.44	0.14	2
Ours	0.45 M	1.22	0.03	1

**Table 5 sensors-24-03070-t005:** Loss ablation experiments. The best results are indicated in red. “w/o” indicates that the corresponding item is not added to the experiment.

Methods	PSNR	SSIM	UIQM
w/o SSIM loss	25.67 ± 2.52	0.75 ± 0.08	2.87 ± 0.36
w/o VGG loss	25.43 ± 2.49	0.77 ± 0.07	2.91 ± 0.25
w/o MSE loss	25.83 ± 2.86	0.78 ± 0.07	2.90 ± 0.31
Ours	27.50 ± 2.78	0.83 ± 0.14	2.93 ± 0.17

**Table 6 sensors-24-03070-t006:** Attention ablation experiments. The best results are indicated in red. “w/o” indicates that the corresponding item is not added to the experiment.

Methods	UFO 120			EUVP		
	PSNR	SSIM	UIQM	PSNR	SSIM	UIQM
Ours w/o SA	24.57 ± 1.78	0.70 ± 0.06	2.80 ± 0.22	26.78 ± 2.53	0.80 ± 0.07	2.89 ± 0.23
Ours w/o CA	24.69 ± 2.32	0.72 ± 0.05	2.83 ± 0.17	26.83 ± 2.46	0.81 ± 0.05	2.91 ± 0.15
Ours	25.25 ± 2.44	0.73 ± 0.08	2.85 ± 0.19	27.50 ± 2.78	0.83 ± 0.04	2.93 ± 0.17

## Data Availability

Data are contained within the article.

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
