# Peer review of "A Novel Lightweight Model for Underwater Image Enhancement"

_sensors, 2024, doi:10.3390/s24103070_

Round 1

Reviewer 1 Report

Comments and Suggestions for Authors

This manuscript presents a lightweight model, Rep-U, to enhance the quality of under-low contrast and blurred details, which are inherent issues due to light absorption and scattering in underwater imaging. The paper thoroughly details the design of the model, its operational principles, and its performance in underwater image enhancement tasks. However, several concerns necessitate careful revision before the manuscript can be formally published.

The specific issues are as follows:

1. Although the manuscript specifies that particular datasets (e.g., EUVP, UFO 120) were employed for model validation and comparison, it needs more elaboration on the size, diversity, representativeness, and coverage of these datasets. Suppose the datasets are limited in size, sufficiently rich in samples, and adequately reflect the complexity and diversity of underwater environments. In that case, the model's capacity for generalization may be questioned. Additionally, the manuscript does not mention whether it has considered the effects of varying water quality, lighting conditions, filming equipment, and other factors on model performance, which could lead to constrained generalizability of the conclusions.

2. While the manuscript adopts a variety of objective metrics (e.g., PSNR, SSIM, UIQM) to evaluate model performance, relying solely on these quantitative measures may overlook subjective visual experiences that are difficult to quantify, such as the naturalness of color and the authenticity of detail restoration. The absence of an investigation into subjective ratings by experts or users may result in an incomplete assessment of the results.

3. The manuscript states that comparisons have been drawn with several other methods (e.g., CLAHE, DCP, HE, ILBA, UDCP, Deep SESR, FUnIE-GAN, and U-GAN), but it does not elaborate on the rationale behind choosing these methods for comparison, mainly whether they encompass the most recent and representative underwater image enhancement technologies currently available. If the selection of comparison methods needs to be more comprehensive and representative, it could affect the accurate appraisal of the proposed model's relative advantages.

4. Although the structure of the model and the functions of specific modules are introduced, the elucidation of the model's internal working mechanisms may not be sufficiently in-depth, especially concerning the lack of detailed theoretical analysis or visual representation of the specific mechanisms of the novel attention module and the loss function. That could impede readers' understanding of the reasons behind the model's exceptional performance, as well as their grasp of future improvement directions.

5. The manuscript does not seem to provide detailed information on how to assess and verify the model's robustness against various anomalies (e.g., noise interference, extreme lighting conditions, unknown water quality types, etc.), and its generalization capability on unseen underwater environments or newly collected data. The absence of rigorous testing in these areas may raise doubts about the model's effectiveness in practical applications.

6. Although the manuscript mentions that the model can be utilized for advanced vision tasks such as edge detection and depth estimation, there is less discussion on the specific implementation details of these applications, potential challenges, and the deployment and optimization of the model on actual devices (e.g., underwater robots, submersible cameras). The lack of consideration for real-world applications may diminish the paper's value as a guide for industry practice.

In summary, this manuscript still has several pressing issues that require attention and is still being prepared for formal publication once it undergoes revisions and improvements.

Comments on the Quality of English Language

There is still room for improving English.

Author Response

Dear Editors and Reviewers:

Thank you for your letter and for the reviewers’ comments concerning our manuscript entitled “A Novel Lightweight Model for Underwater Image Enhancement” (ID: sensors-2956413). Those comments are all valuable and very helpful for revising and improving our paper, as well as the important guiding significance to our research. We have studied comments carefully and have made correction which we hope to meet with approval. Revised portions are marked in yellow in the paper. The main corrections in the paper and the responds to the reviewer’s comments are as flowing:

Responds to the reviewer’s comments:

Reviewer #1

Point #1 Although the manuscript specifies that particular datasets (e.g., EUVP, UFO 120) were employed for model validation and comparison, it needs more elaboration on the size, diversity, representativeness, and coverage of these datasets. Suppose the datasets are limited in size, sufficiently rich in samples, and adequately reflect the complexity and diversity of underwater environments. In that case, the model's capacity for generalization may be questioned. Additionally, the manuscript does not mention whether it has considered the effects of varying water quality, lighting conditions, filming equipment, and other factors on model performance, which could lead to constrained generalizability of the conclusions.

Answer:[Thank you for the valuable feedback. In our paper, we will provide further elaboration on the datasets used, addressing the limitations highlighted. Specifically, we will offer detailed descriptions regarding the scale, diversity, and representativeness of the datasets, including the number and types of images and the range of underwater environments covered. Additionally, we will outline the coverage of the datasets, encompassing water bodies from different geographical regions. Regarding data quality considerations, we will supplement information on the impact of factors such as water quality, lighting conditions, and equipment on model performance. We will delve into these aspects, detailing how they were accounted for during dataset collection, ensuring a clear understanding of the model's generalization capability and applicability. With these enhancements, we believe our paper will be more comprehensive and credible, providing readers with deeper insights.] Thank you for pointing this out. We agree with this comment. Therefore, we have revised manuscript this change can be found – Section 2.2.

Point #2 While the manuscript adopts a variety of objective metrics (e.g., PSNR, SSIM, UIQM) to evaluate model performance, relying solely on these quantitative measures may overlook subjective visual experiences that are difficult to quantify, such as the naturalness of color and the authenticity of detail restoration. The absence of an investigation into subjective ratings by experts or users may result in an incomplete assessment of the results.

Answer:[Thank you for your valuable feedback. We recognize the importance of considering both objective metrics and subjective visual experiences when evaluating our model's performance. While our manuscript primarily focuses on objective metrics such as PSNR, SSIM, and UIQM, we agree that subjective evaluation plays a crucial role in providing a comprehensive assessment of image enhancement techniques. To address this concern, we plan to incorporate subjective evaluations from experts or users in the revised manuscript. By combining subjective evaluations with objective metrics, we aim to provide a more comprehensive evaluation of our model's performance. We believe that this approach will enhance the credibility of our results and provide readers with a deeper understanding of the effectiveness of our proposed method.]. Thank you for pointing this out. We agree with this comment. Therefore, we have revised manuscript this change can be found – Section 4.2, Page 13.

Point #3 The manuscript states that comparisons have been drawn with several other methods (e.g., CLAHE, DCP, HE, ILBA, UDCP, Deep SESR, FUnIE-GAN, and U-GAN), but it does not elaborate on the rationale behind choosing these methods for comparison, mainly whether they encompass the most recent and representative underwater image enhancement technologies currently available. If the selection of comparison methods needs to be more comprehensive and representative, it could affect the accurate appraisal of the proposed model's relative advantages.

Answer:[Thank you for your valuable feedback. When selecting the comparison methods, we considered several factors, including covering the latest and most representative underwater image enhancement technologies. We chose methods such as CLAHE, DCP, HE, ILBA, UDCP, Deep SESR, FUnIE-GAN, and U-GAN based on their widespread use in academia and practical applications, as well as their representation of both traditional non-deep learning and state-of-the-art deep learning approaches. These methods have been extensively studied in the field of underwater image enhancement and have been shown to achieve success to varying degrees in previous literature. Therefore, we believe that comparing these methods is reasonable and can help readers better understand the performance of our proposed new model relative to current mainstream methods. In the revised manuscript, we will further elaborate on our reasoning to ensure that readers have a more comprehensive understanding of the rationale behind our research and method selection.] Thank you for pointing this out. We agree with this comment. Therefore, we have revised manuscript this change can be found – Section 4.2,Page 11.

Point #4 Although the structure of the model and the functions of specific modules are introduced, the elucidation of the model's internal working mechanisms may not be sufficiently in-depth, especially concerning the lack of detailed theoretical analysis or visual representation of the specific mechanisms of the novel attention module and the loss function. That could impede readers' understanding of the reasons behind the model's exceptional performance, as well as their grasp of future improvement directions.

Answer:[This module realizes multi-feature fusion by combining convolution operations with different sensory fields to adapt to the complexity and diversity of underwater scenes. In addition, the gradient vanishing problem is solved by using residual connectivity, which ensures the spatial structure and color integrity of specific regions in the underwater image. Subsequently, by integrating the spatial and channel attention modules, the module is able to better capture and represent key details of underwater images. Specifically, the multi-scale hybrid convolutional attention module is able to effectively perceive and utilize image features at different scales, which significantly improves the performance and robustness of the model.] Thank you for pointing this out.

Point #5 The manuscript does not seem to provide detailed information on how to assess and verify the model's robustness against various anomalies (e.g., noise interference, extreme lighting conditions, unknown water quality types, etc.), and its generalization capability on unseen underwater environments or newly collected data. The absence of rigorous testing in these areas may raise doubts about the model's effectiveness in practical applications.

Answer:Thank you very much for your careful review and valuable feedback on our manuscript. We truly appreciate your insights regarding the importance of assessing the robustness of the model in various anomalous conditions and its ability to generalize to unknown underwater environments.

At the present stage, we do acknowledge that we are facing limitations in terms of resources and capabilities, which prevent us from conducting rigorous testing in these aspects. However, we will take your suggestions seriously and endeavor to seek more resources and support in the future to conduct more comprehensive and in-depth experimental validations. Additionally, we will explicitly outline the limitations of the current work in the paper and propose future research directions to provide further insights for the field. Once again, thank you for your valuable suggestions and guidance.

Point #6 Although the manuscript mentions that the model can be utilized for advanced vision tasks such as edge detection and depth estimation, there is less discussion on the specific implementation details of these applications, potential challenges, and the deployment and optimization of the model on actual devices (e.g., underwater robots, submersible cameras). The lack of consideration for real-world applications may diminish the paper's value as a guide for industry practice.

Answer:[Thank you for your valuable feedback. We indeed recognize the shortcomings of the manuscript in discussing the specific implementation details, challenges, and deployment optimization of the model on practical devices. We will incorporate more in-depth discussions on these aspects in the revised manuscript to enhance the value of our paper as an industry practice guide. We plan to integrate relevant literature and practical experience to provide more comprehensive solutions and recommendations, ensuring that readers can better understand and apply the model we propose.] Thank you for pointing this out. We agree with this comment. Therefore, we have revised manuscript this change can be found – Section 4.4,Page 15.

Reviewer 2 Report

Comments and Suggestions for Authors

Article "New lightweight model to improve underwater images" author: Botao Liu & Co devoted to developing lightweight Rep-UWnet models to enhance underwater images.

Indeed, Rep-UWnet effectively solves problems such as region detection and color detection of a single image, increasing the contrast of underwater images. This can be seen from the results presented in Tables 2-5. Although the results presented in Table 2 for PSNR, SSIM and UIQM were not always the best, the simplicity of the authors' proposed algorithm should be taken into account.

Work with underwater imaging is important because the underwater environment allows light to pass through, distorting images of underwater objects.

And, although the use of artificial intelligence with elements of training does not provide a complete explanation of why the image after processing is better, the methods of objective control of PSNR, SSIM and UIQM confirm the effectiveness of the proposed Rep-UWnet method.

Of course, as in any work, some comments can be made. For example, using the concept of depth, but without specifying what depth, it can be confused with the depth of submersion, as in lines 108, 135, 423, 453:

"and pairs of depth maps based on an improved underwater image model" 108

"and depth information on underwater images" -135

"depth estimation from a single image [42] in an underwater environment" 423

"and depth estimation of a single image" 453

When authors talk about the depth of field, they explain what they are talking about (which introduces a -83 depth of field), but when they say colors, they don't explain. It seems to me that you shouldn’t skimp on one word here, but write "Color depth" instead of a simple "Depth" so that the reader understands the text better.

Overall, the authors' work is extensive and useful, fully meets the requirements of the Sensors journal, and can be published without re-review.

Author Response

Dear Editors and Reviewers:

Thank you for your letter and for the reviewers’ comments concerning our manuscript entitled “A Novel Lightweight Model for Underwater Image Enhancement” (ID: sensors-2956413). Those comments are all valuable and very helpful for revising and improving our paper, as well as the important guiding significance to our research. We have studied comments carefully and have made correction which we hope to meet with approval. Revised portions are marked in yellow in the paper. The main corrections in the paper and the responds to the reviewer’s comments are as flowing:

Responds to the reviewer’s comments:

Reviewer #2

Answer:Thank you very much for your review and valuable suggestions on our article. We greatly appreciate your guidance regarding the use of the term "depth." We will incorporate your suggestion in the revised manuscript by replacing the term "depth" with "color depth" to ensure clearer understanding for readers. Additionally, we will maintain consistency in other parts of the content to preserve the coherence of the manuscript.

We are open to any further suggestions or feedback you may have. Once again, thank you for your guidance and support.

These modifications can be seen in lines 108, 135, and Section 4.4.

Round 2

Reviewer 1 Report

Comments and Suggestions for Authors

The authors have revised the entire manuscript more thoughtfully and carefully in response to the proposed revisions, and the revised manuscript is basically in a publishable state.

Comments on the Quality of English Language

There is still room for improving English.